# Evaluation of Dry Eye Treatment with Sodium Hyaluronate- and Dexpanthenol-Containing Eye Drops on Ocular Surface Improvement after Cataract Surgery

**DOI:** 10.3390/diagnostics14111097

**Published:** 2024-05-25

**Authors:** Maximilian K. Köppe, Mustafa K. Hallak, Annette L. Stengele, Ramin Khoramnia, Gerd U. Auffarth

**Affiliations:** International Vision Correction Research Centre (IVCRC), University Eye Clinic Heidelberg, 69120 Heidelberg, Germany; maximilian.koeppe@med.uni-heidelberg.de (M.K.K.); mustafa.hallak@med.uni-heidelberg.de (M.K.H.); stengele@augen-darmstadt.de (A.L.S.); ramin.khoramnia@med.uni-heidelberg.de (R.K.)

**Keywords:** dry eye, ocular surface, cornea

## Abstract

Background: To clinically evaluate how dry eye symptoms in preoperatively diagnosed dry eye patients change with the use of sodium hyaluronate- and dexpanthenol-containing eye drops (HYLO CARE (HC), URSAPHARM Arzneimittel GmbH, Saarbruecken, Germany) after cataract surgery. The aim of the study was not to compare different eye drops but to implement standard treatment in patients with dry eye undergoing cataract surgery. The impact of treatment was evaluated using Symptom Assessment Tools for Dry Eye. Methods: In this prospective, single-center, open-label clinical trial, 49 patients undergoing cataract surgery were included who showed signs and symptoms of dry eye disease assessed by the Symptom Assessment in Dry Eye (Visual Analogue Scale (VAS)) questionnaire, Ocular Surface Disease Index (OSDI), and fluorescein tear break-up Time (TBUT). Patients were instructed to apply HC three to four times a day for 5 weeks in the operated eye in addition to the standard postoperative topical anti-inflammatory regimen. The primary endpoint was the change in TBUT. Secondary endpoints were the assessment of the subjective symptoms (VAS), corrected distance visual acuity (CDVA), and slit-lamp examination including the corneal staining score, Schirmer test, and intraocular pressure. Results: At 5 weeks after operation, the mean TBUT increased from 6.42 ± 1.57 s (s) to 7.81 ± 1.83 s in the per-protocol (PP) population (*p* > 0.001) and from 6.33 ± 1.64 s to 7.71 ± 2.05 s in the intention-to-treat (ITT) population (*p* < 0.001). There was a statistically significant decrease in all scores (*p* < 0.05) from the VAS questionnaire except for the tearing score (*p* = 0.062) at 5 weeks after operation. The mean total corneal staining score also decreased statistically significantly from 8.85 ± 2.49 before operation to 5.61 ± 3.37 at 5 weeks after operation on a 15-point scale. Conclusions: Controlled standardized dry eye treatment (with HC) improved tear film stability, ocular surface defects, and subjective symptoms of dry eye disease in patients 5 weeks after undergoing cataract surgery. Both the patient and physician assessments indicated high efficacy, tolerability, and a reliable safety profile, as indicated by the low number of at least possibly related adverse events (AE), suggesting its beneficial role in the postoperative management of the ocular surface (OS) in patients with dry eye symptoms prior to and after cataract surgery.

## 1. Introduction

Cataract surgery can have detrimental effects on the ocular surface, not only by causing and exacerbating dry eye disease (DED) symptoms but also by increasing the risk of infection after cataract surgery [1]. Dasgupta and Gupta found that 100% of patients showed abnormalities in TBUT, the Schirmer I test, and DED symptomatology 12 weeks after cataract surgery [2]. Given the high volume of cataract surgeries performed per year in the developed world, DED after cataract surgery thus demonstrates a frequently encountered issue in everyday practice. Disturbances of the tear film were observed after cataract surgery using the phacoemulsification technique or femtosecond laser-assisted cataract surgery (FLACS) [3] and even after manual small incision cataract surgery (SICS) [1].

DED affects the tear film and the ocular surface (OS) and may lead to serious damage to the cornea and conjunctiva if not adequately treated. The multifactorial nature of dry eye is described by the Dry Eye Workshop (DEWS II) as a disease in which the loss of homeostasis of the tear film is the central pathophysiological concept [4]. 

The most common DED therapy is the substitution of the tear film [5]. DEWS II states that “Tear replacement with ocular lubricants is traditionally considered a mainstay of DED therapy” [6]. 

Ophthalmic preparations containing hyaluronic acid have been reported to be used successfully in various pathological disorders of the physiology and morphology of the eye [7,8]. Stuart et al. showed that the subjective well-being of the patient improved dramatically with the administration of eye drops containing sodium hyaluronate [9]. Dexpanthenol is a hygroscopic substance with a high water-binding capacity. As clinical studies have shown, the topical application of products containing dexpanthenol increases the water content of the epithelia of the skin [10]. Adequate moisture is necessary to keep the skin and mucous membranes healthy. Wound healing is improved under moist conditions [11]. Due to these properties, dexpanthenol also supports the moistening of the surface of the eye with sodium hyaluronate. The natural balance of humidification is restored, and the healing of epithelial damage is supported [10,11,12,13].

The well-established tear supplement HYLO CARE (HC) (URSAPHARM Arzneimittel GmbH, Saarbruecken, Germany) eye drops contain sodium hyaluronate, dexpanthenol, citrate buffer, and water. The supplement is free of phosphates and preservatives, does not contain any emulsifiers, and is certified for postoperative application.

The aim of this prospective, single-center, open-label clinical trial was to evaluate the efficacy and tolerability of HC on patients with DED undergoing cataract surgery using standard symptom assessment tools for dry eye.

## 2. Patients and Methods

In this prospective, single-center, open-label clinical trial, 49 patients were included and examined preoperatively and 1 day, 1 week, and 5 weeks postoperatively. The trial adhered to the German Act on Medical Devices (MPG) § 23b [14] and the EN ISO 14155: 2012-01 [15] clinical trial on medical devices for use in humans. The trial was carried out according to the trial protocol and good clinical practice (GCP) and followed the Declaration of Helsinki and the laws of the Federal Republic of Germany. The trial was submitted to the ethics committee of the medical faculty of the University of Heidelberg for review and professional advice prior to the enrollment of the first patient receiving approval for the trial. All participants were comprehensively informed about the study and provided their informed consent.

The main inclusion criteria were male and female adults with the presence of age-appropriate cataract (uni- or bilateral) for which standard cataract surgery was expected and preoperative subjective complaints indicating dry eye (foreign body sensation, dryness, burning, tearing, or tired eyes) for at least 3 months with a Symptom Assessment in Dry Eye Visual Analog Scale (VAS) questionnaire score of ≥2/10, an OSDI score of ≥16, and a fluorescein TBUT of ≤10 s. The participant had to have stable topical or systemic therapy for ≥4 weeks, be willing to participate in the trial, and be willing and able to fulfill the requirements of the trial protocol. The exclusion criteria were dry eye due to systemic disease, concomitant medication, malign conditions; the simultaneous use of other ophthalmic therapeutics; ocular or systemic pathologies that may have an influence on the postoperative irritation (e.g., acute viral or bacterial inflammation of the conjunctiva/cornea or chronic inflammatory/infectious uveitis); a history of ocular surgery or punctum plugs during the past 3 months; malposition of the eyelids and/or lagophthalmos; contact lens use up to 3 months before surgery; hypersensitivity against one of the ingredients of HC; and concurrent pregnancy or lactation.

Upon inclusion in the study, all patients were instructed to place one drop of HC in the conjunctival sac of the study eye that underwent surgery 3–4 times daily in addition to the standard postoperative topical treatment (IsoptoMax eye drops; (Novartis Pharma GmbH, Nuremberg, Germany) for 2 weeks and IsoptoMax eye ointment (Novartis Pharma GmbH, Nuremberg, Germany) (1 × 1/night for 1 week). No application of any eye drops was allowed up to 2 h before an examination.

### 2.1. Preoperative and Postoperative Examination

This trial included four visits. The preoperative examination was before cataract surgery (D0). The follow-up examinations were defined in the trial protocol as day 1 (D1) and day 7 (D7) ± 3 days (=1 week) after cataract surgery. The final examination was scheduled for day 35 (D35) ± 5 days (=5 weeks) after cataract surgery. At the preoperative visit, all patients were asked to complete the VAS and OSDI questionnaires to evaluate the severity and frequency of symptoms of dry eye disease (foreign body sensation, dryness, burning, tearing, and tired eyes) [16]. Preoperatively, values were expressed on a VAS from 0 to 10. At 1 and 5 weeks after operation, subjective changes in these values were further evaluated via VAS, and they were transformed into a decimal scale using the percentage change from baseline. OSDI scores were used to assess the severity of dry eye symptoms according to patients’ subjective feelings and were categorized as normal (0–12 points), mild (13–22 points), moderate (23–32 points), or severe (33–100 points) [16,17]. In this trial, the OSDI was used preoperatively to verify eligibility for the study, but it was not used as a measurement postoperatively.

During all study visits, participants underwent a complete ophthalmologic assessment, including slit-lamp examination, visual acuity assessment, TBUT measurement, and Goldmann applanation tonometry to measure intraocular pressure. Corneal staining was graded according to the Oxford grading scale (max. 15 points) [18]. At the preoperative and 5-week postoperative visits, the Schirmer test was performed with paper strips inserted into the lower eyelid pouch for 5 min without anesthesia (Schirmer I) [19].

The participants were closely monitored over the course of the trial and carefully examined for any adverse events.

### 2.2. Demographic Data

The trial results refer to different analysis populations. The per-protocol (PP) analysis was defined as the analysis including all patients who applied the trial treatment, appeared for all follow-up visits, and had no serious investigation plan deviations.

The intention-to-treat (ITT) analysis was defined as the analysis including all patients who received the investigational drug and had at least one follow-up appointment after the use of the trial treatment. The primary objective, TBUT change, was evaluated in the PP population. All other variables refer to the ITT population. Overall, 67 patients (67 eyes) were screened for this trial. A total of 18 patients dropped out before treatment due to different reasons (e.g., cataract surgery was canceled, complications after surgery, or no reason given) and were excluded from the final data analysis. Also, the pandemic situation during the course of the trial contributed to the high drop-out rate. Thus, a total of 49 patients were treated with HC and included in the safety analysis. A total of 10 of these 49 patients were not included in the efficacy assessments (i.e., the PP and ITT populations), as five patients who were enrolled in the trial and treated with HC were discovered to have violated the defined inclusion/exclusion criteria at the start of the trial, and five patients terminated the trial prematurely after the visit on D1 (*n* = 1) or D7 (*n* = 4) without giving a reason.

### 2.3. Sample Size Calculation

For sample size calculation, peer-reviewed publications were used to define potential means and standard deviations of the mean for TBUT values [19,20]. The sample size calculation was performed according to Dupont and Plummer [21]. A total of 35 patients were needed to detect a difference in preoperative TBUT (D0) and TBUT 35 days after cataract surgery, with an expected preoperative TBUT mean of 6.5 s and an expected postoperative TBUT mean (on D35) of 9.0 s with a standard deviation (SD) of 4.0, a statistical power of 95%, and a significance level of 0.05. A total of 40 patients had to be recruited to compensate for a dropout rate of 10–15%.

### 2.4. Statistical Analysis

Data analysis was performed using SPSS Statistics Software Package Version 19.0 for Windows (SPSS, Chicago, IL, USA). The descriptive statistics of all test parameters at the four examination visits (D0, D1, D7, and D35) contained the sample size, mean value, standard deviation, minimum, maximum, and median. Categorical data are expressed as frequencies and percentages. Clinical progression parameters were analyzed and displayed as intraindividual differences between preoperative D0 and postoperative D35 using the Wilcoxon signed-rank test. Paired categorical data were evaluated with the exact McNemar test. Preoperatively, the scores of the patient questionnaire were expressed on a VAS from 0 to 10. Postoperatively, subjective changes in these values were further evaluated via VAS. For data analysis, all scores were transformed into a decimal scale by measuring the distance between 0 and the mark in millimeters. For accurate statistical analysis of the visual acuity outcomes, decimal values were transformed into logMAR notation. For all monocular variables, only one eye per patient was included in the data analysis. If both eyes had to undergo cataract surgery within 1 or 2 weeks (due to bilateral cataracts), HC had to be applied to one eye exclusively. Only the results of this eye were documented. For all statistical tests, a *p*-value of less than 0.05 was considered statistically significant.

## 3. Results

A total of 39 patients constituted the ITT population. For the PP analysis, 3 of these 39 patients had to be excluded (one patient discontinued the use of HC prematurely due to intolerance, and two patients had visits later than defined in the trial protocol). Thus, 36 patients were included in the per-protocol population that was used to assess the primary objective of this trial, tear break-up time.

Table 1 shows mean values of the TBUT, CDVA, Schirmer test, corneal staining score, and intraocular pressure (IOP) at given visit dates.

### 3.1. Fluorescein Tear Break-Up Time

#### 3.1.1. PP Population

The TBUT results of the PP population (*n* = 36) are presented in Figure 1. The mean TBUT (±SD) before surgery (D0) was 6.42 ± 1.57 s. On D35, TBUT increased to 7.81 ± 1.83 s. TBUT on postoperative D35 was significantly increased compared with the preoperative visit on D0 (*p* < 0.001).

#### 3.1.2. ITT Population

The TBUT results of the ITT population (D0 to D7: *n* = 39; D35: *n* = 38) are presented in Figure 2. The Mean TBUT (±SD) before surgery (D0) was 6.33 ± 1.64 s. On D35, the TBUT increased to 7.71 ± 2.05 s (*p* < 0.001).

### 3.2. Corrected Distance Visual Acuity (CDVA)

There was a statistically significant improvement (*p* < 0.001) in mean CDVA values (logMAR), from 0.51 ± 0.35 before operation to 0.09 ± 0.20 at 5 weeks after surgery.

### 3.3. Schirmer Test

The mean values (±SD) of Schirmer’s test were 10.24 ± 9.44 mm before surgery (D0) and 10.92 ± 7.94 mm at the final examination. There was no statistical difference in Schirmer’s test outcomes between D0 and D35 (*p* = 0.309).

### 3.4. Corneal Staining

The mean corneal staining score showed a statistically significant improvement (*p* < 0.001), from 8.85 ± 2.49 before operation to 5.61 ± 3.37 after 5 weeks on a 15-point scale.

### 3.5. Intraocular Pressure (IOP)

IOP showed no significant change between preoperative values and values after 5 weeks of treatment (*p* = 0.814).

### 3.6. Visual Analogue Scale (VAS) Questionnaire

Table 2 shows the scores given by the patients at the preoperative and 1-week and 5-week postoperative visits. With respect to the baseline values, all parameters except for the tearing score showed a statistically significant improvement (*p* < 0.05) at the 5-week visit.

### 3.7. Subjective Assessment of Tolerability and Effectiveness

Patients assessed the tolerability of HC after 1 week (D7) and 5 weeks of treatment (D35), and the physician assessed it after 5 weeks of treatment (D35) as either ‘impeccable’, ‘acceptable’, or ‘not acceptable (Figure 3). The majority of patients assessed the tolerability of HC as ‘impeccable’ at both time points, with 85% (*n* = 33) on D7 and 89% (*n* = 34) on D35. By contrast, 15% of patients (*n* = 6) on D7 and 11% of patients (*n* = 4) on D35 rated the tolerability as ‘acceptable’. The physician assessed the tolerability in 95% of the patients (*n* = 36) as ‘impeccable’ and that in 5% of the patients (*n* = 2) as ‘acceptable’ on D35. No patient or physician rated the product as ‘not acceptable’. (Figure 1). The efficacy of HC was assessed after 5 weeks of treatment (D35) by the physician as either ‘disappointing’, ‘sufficient’, ‘good’, or ‘very good’. In 97.4% of cases (*n* = 37), the efficacy was assessed as ‘very good’ or ‘good’. For only one patient, the efficacy was assessed as ‘sufficient’ by the physician. No cases of ‘disappointing’ efficacy were observed.

### 3.8. Adverse Events

During the trial, 23 non-serious adverse events (AEs) were observed in 15 patients (safety population: *n* = 49) as listed in Table 3. No serious AEs were observed. Out of these 23 non-serious AEs, one AE, ‘burning right eye’, was suspected to be caused by HC with at least ‘possible’ causality. The patient discontinued the treatment with HC because of the AE and was not recovered at the time of the AE report. Most of the AEs were related to the cataract surgery.

## 4. Discussion

Evidence of ocular surface dysfunction is frequently observed in patients undergoing cataract surgery, often undetected until presentation [22]. Sacchi et al. (2009) conducted a study documenting ocular changes during cataract surgery [23]. They noted that 71% of patients experienced a notable reduction in the sub-basal nerve plexus fibers, with 60% showing a shift in nerve plexus morphology from regular to irregular (reticular-like aspect) after surgery.

Ishrat et al. (2019) investigated the occurrence and pattern of dry eye disease (DED) following phacoemulsification and manual small incision cataract surgeries (SICS). None of the patients exhibited DED symptoms preoperatively, yet 42% reported dry eye symptoms by day 7 after surgery [1]. This trial encompassed patients meeting dry eye criteria before cataract surgery. After surgery, patients received treatment from day 1 to day 35 with HC along with standard anti-inflammatory/antibiotic topical therapy from the treating center. Objective and subjective parameters describing tear film and ocular surface of patients were meticulously documented in this clinical trial.

Cataract surgeries in this trial proceeded smoothly for all patients, except for one instance of capsule rupture during surgery, resulting in exclusion from further participation. As anticipated, corrected distance visual acuity (measured in logMAR) significantly improved after cataract surgery, from a mean of 0.51 ± 0.35 preoperatively to 0.09 ± 0.20 on day 35 (*p* < 0.001). Visual acuity and intraocular pressure measurements, which remained relatively stable over time, served as pivotal ophthalmological safety indicators.

The primary efficacy parameter TBUT exhibited a significant increase in the PP population, from 6.42 ± 1.57 s before operation to 7.81 ± 1.83 s after 5 weeks of treatment. Hence, the null hypothesis for TBUT (no difference between preoperative and day-35 values) was rejected. TBUT not only remained stable but also showed a slight improvement, achieving the main objective of preventing aggravated ocular surface dysfunction after cataract surgery in patients with DED. In accordance with Sullivan (2010), patients with a TBUT of 6.1 ± 4.9 are categorized as experiencing mild to moderate DED severity. As a comparison, normal values are considered to be 11.8 ± 6.4 [24].

The patients in the present trial exhibited lower TBUT values pre-surgery compared with those reported by Ishrat et al. (2019), with normal preoperative TBUT values of 15.8 s (day 0), 8.7 s on day 7, and 11.2 s 4 weeks after surgery. These patients did not receive topical DED medication and did not experience preoperative DED [1]. TBUT decreased in both studies after surgery but improved over time. However, in the trial described herein, TBUT was higher after 5 weeks of treatment compared with baseline, a result not achieved for patients without tear substitutes as presented by Ishrat et al. [1].

The implementation of strict inclusion criteria represents a strength of the study and improves the validity of the present study. OSDI serves as a crucial marker in DED and was utilized in this trial to assess the functional status of the ocular surface of patients with cataracts before surgery on the basis of subjective patient impressions. At baseline, all 39 patients exhibited severe OSDI (score ≥ 33), with a mean of 65.28 ± 15.46. The total corneal staining score (ranging from 0 to 15) significantly decreased from 8.85 ± 2.49 pre-surgery to 5.61 ± 3.37 (Δ3.24) on day 35 (*p* < 0.001), indicating postoperative improvement similar to TBUT. The reduction in total corneal staining score clearly signifies an improvement in ocular surface condition, corroborating subjective symptoms reported by patients via the SANDE patient questionnaire, which documented the frequency and severity of symptoms, such as tearing, burning, foreign body sensation, dryness, and tired eye sensation preoperatively and after 1 and 5 weeks of HC treatment. Except for tearing, all subjective signs of DED exhibited significant improvement throughout the trial, with a clear tendency for improvement in the tearing score as well. Schirmer’s test values showed no statistically significant change between day 0 and day 35, indicating that postoperative HC application did not influence tear quantity as expected. The positive objective results and subjective patient impressions in this trial align with assessments of tolerability and efficacy by both patients and physicians. The vast majority of patients rated the tolerability of HC as ‘impeccable’. No instances of ‘unacceptable’ tolerability were reported. Physician-rated efficacy of HC was predominantly ‘very good’ or ‘good’ in 37 patients, with one patient deemed as ‘sufficient’.

During the trial, a total of 23 non-serious adverse events (AEs) in 15 patients were reported, with only one AE, ‘burning right eye’, classified as possibly related to HC use. This AE, although not serious, led to treatment discontinuation. Hypersensitive reactions such as burning are well-documented. No risks associated with HC application post-cataract surgery were identified in this trial. No emergency events occurred, and overall safety was deemed satisfactory. HC was well-tolerated both locally and systemically.

The positive effects of HC on TBUT and corneal staining may be attributed to the formation of an even and long-lasting lubricating film on the eye surface, supported by wound healing due to dexpanthenol. The results presented herein demonstrate that after cataract surgery, a 35-day regimen of HC application four times daily can ameliorate signs and symptoms of dry eye in patients.

## 5. Conclusions

HC proved to be effective in treating patients with symptoms of DED after cataract surgery. Both the patient and physician assessments indicated high tolerability and a reliable safety profile, suggesting its beneficial role in the postoperative management of the OS in patients with DED.

## Figures and Tables

**Figure 1 diagnostics-14-01097-f001:**
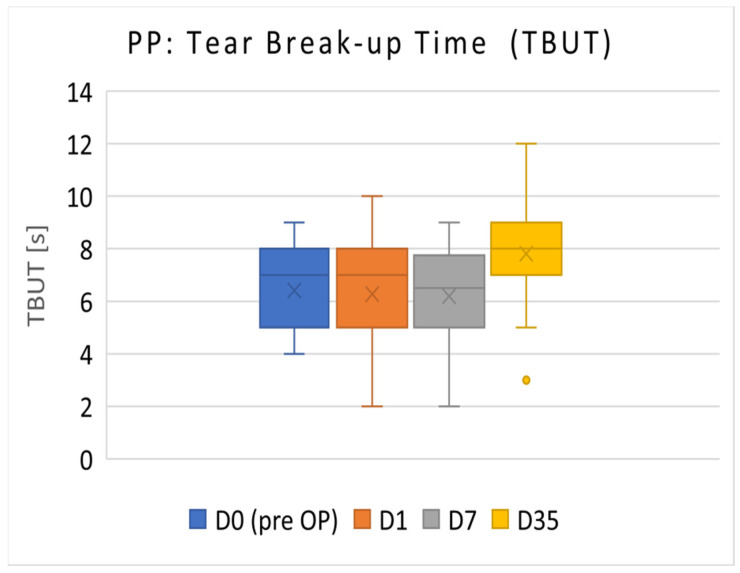
Mean TBUT values of PP population in seconds (primary efficacy). TBUT on postoperative D35 was significantly increased compared with the preoperative visit on D0 (*p* < 0.001).

**Figure 2 diagnostics-14-01097-f002:**
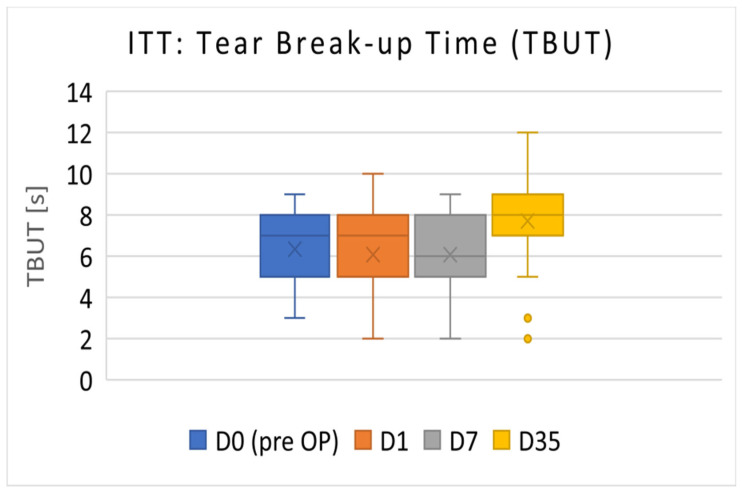
Median TBUT values of ITT population in seconds (primary efficacy). On D35, TBUT increased to 7.71 ± 2.05 s (*p* < 0.001).

**Figure 3 diagnostics-14-01097-f003:**
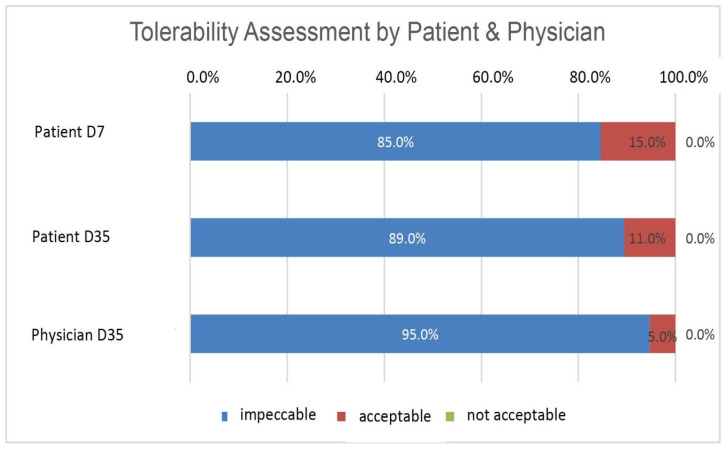
Patient and physician assessment of tolerability (% of patients).

**Table 1 diagnostics-14-01097-t001:** Mean values of the TBUT, CDVA, Schirmer test, corneal staining score, and intraocular pressure (IOP) at given visit dates.

Parameter	Preoperative	1 Day Postoperative	1 Week Postoperative	5 Weeks Postoperative	*p*-Value
TBUT (seconds) (ITT)	6.33 ± 1.64	6.08 ± 1.98	6.08 ± 1.99	7.71 ± 2.05	*p* < 0.001
TBUT (seconds) (PP)	5.42 ± 1.57	6.28 ± 1.91	5.19 ± 1.89	7.81 ± 7.83	*p* < 0.001
CDVA (logMAR)	0.51 ± 0.35	0.25 ± 0.30	0.19 ± 0.28	0.09 ± 0.20	*p* < 0.001
Schirmertest (mm/5 min)	10.24 ± 9.44	-	-	10.92 ± 7.94	*p* < 0.001
Corneal staining	8.85 ± 2.49	8.36 ± 2.59	8.13 ± 3.52	5.61 ± 3.37	*p* < 0.001
IOP (mmHg)	14.45 ± 2.90	15.95 ± 5.18	14.82 ± 2.05	14.15 ± 2.52	*p* < 0.001

**Table 2 diagnostics-14-01097-t002:** Outcomes of VAS Questionnaire.

Parameter	Preoperative	1 Week Postoperative	5 Weeks Postoperative	*p*-Value
Frequency of symptoms	56.00 ± 20.26	39.82 ± 21.81	30.05 ± 21.15	*p* < 0.001
Severity of symptoms	52.67 ± 19.67	40.24 ± 19.28	29.87 ± 2.88	*p* < 0.001
Tearing score	45.56 ± 25.08	43.50 ± 23.13	33.76 ± 23.52	*p* = 0.062
Dryness	51.13 ± 24.55	36.82 ± 18.94	28.42 ± 19.61	*p* < 0.001
Burning eyes	47.03 ± 25.78	36.03 ± 22.02	23.95 ± 21.18	*p* < 0.001
Foreign body sensation	57.15 ± 25.95	42.00 ± 24.95	27.08 ± 23.49	*p* < 0.001
Tired eyes	46.33 ± 21.62	41.79 ± 22.15	31.50 ± 21.93	*p* = 0.011

**Table 3 diagnostics-14-01097-t003:** Adverse events noted during the trial.

Type of Report	Pat. ID	HYLO CARE Suspected of Causing Adverse Event?	Start of Treatment (Date)	Adverse Event Start (Date)	Treatment Continued?	End of Treatment (Date) Due to Adverse Event	Adverse Event (Severity Grade 1–3)
**Initial report**	6	--	05.06.2019	02.07.2019	yes	--	Posterior vitreous body detachment OS (grade 1)
**Initial report**	10	--	17.07.2019	02.08.2019	no	04.08.2019	Pain at OD (grade 1)
**Initial report**	10	--	17.07.2019	17.07.2019	no	04.08.2019	Increased intraocular pressure (grade 1)
**Initial report**	10	--	17.07.2019	02.08.2019	no	04.08.2019	Burning eye (grade 1)
**Initial report**	10	--	17.07.2019	02.08.2019	no	04.08.2019	Itching OD (grade 1)
**Initial report**	10		17.07.2019	23.07.2019	no		Eye irritation (grade 1)
**Initial report**	13	--	14.08.2019	14.08.2019	yes	--	Red eye OS (grade 1)
**Initial report & follow up**	20	--	15.01.2020	15.01.2020	--	--	Increased intraocular pressure OD
**Initial report**	**20**	**yes**	**15.01.2020**	**07.02.2020**	**no**	**12.02.2020**	**Burning OD**
**Initial report**	22	--	10.01.2020	11.02.2020	yes	11.02.2020	Cystoidal macular edema (grade 1)
**Initial report**	26	no	06.02.2020	13.02.2020	yes	--	Postoperative intraocular pressure derailment (grade 1)
**Initial report**	28	--	25.02.2020	25.02.2020	--	--	OD increased ocular pressure
**Initial report**	28		25.02.2020	28.02.2020	yes	--	OD burning after Isoptomax eye drops
**Initial report**	29	--	29.01.2020	29.01.2020	--	--	Burning inside lower lid
**Initial report**	32	--	13.02.2020	13.02.2020	yes	--	Intraocular pressure increase OD (grade 1)
**Initial report**	33	--	04.03.2020	19.03.2020	--	02.04.2020	Redness at OD (anamnestic)
**Initial report**	42	--	04.08.2020	08.08.2020	yes	--	Descemet folds
**Initial report**	42	--	04.08.2020	08.08.2020	yes	--	Loosened macula
**Initial report**	43	--	14.08.2020	04.09.2020	yes	--	Candida albicans
**Initial report**	48	--	01.10.2020	06.10.2020	yes	--	Makular edema (grade 1)
**Initial report**	59	--	16.02.2021	17.03.2021	yes	--	Uveitis anterior (grade 1)
**Initial report**	65	--	04.05.2021	04.05.2021	yes	--	Tensio decompensation
**Initial report**	65	--	04.05.2021	04.05.2021	yes	--	Capsular defect, posterior

## Data Availability

The datasets used and/or analyzed during the current study are available from the corresponding author upon reasonable request.

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
