# Peer review of "Evaluation of Dry Eye Treatment with Sodium Hyaluronate- and Dexpanthenol-Containing Eye Drops on Ocular Surface Improvement after Cataract Surgery"

_diagnostics, 2024, doi:10.3390/diagnostics14111097_

Round 1

Reviewer 1 Report (New Reviewer)

Comments and Suggestions for Authors

This is a very interesting research article concerning the assessment of dry eye treatment in patients undergoing cataract surgery using eye drops containing sodium hyaluronate and dexpanthenol (HYLO CARE, URSAPHARM Arzneimittel GmbH, Saarbruecken, Germany).

This article is well-organized and well written, and the authors clearly specified the aim of their study and the endpoints.

Just some few points to improve the manuscript:

1) The authors should specify the abbreviations "PP" and "ITT" in the Abstract.

2) The authors should also specify "OS" in the Introduction section.

3) I suggest to include all the "Results" in one paragraph, except for "Subjective Assessment of Tolerability and Effectiveness" and "Adverse Events" that could remain as single paragraphs.

4) The Discussion section appears too long and looks like a "wall of text", also with too small sentences, especially in the last part of the section. I suggest to revise all this section.

Author Response

Thank you very much for your invaluable and insightful report. 

We ammended the manuscript to your feedback bullet points accordingly. 

Thank you very much! 

Reviewer 2 Report (New Reviewer)

Comments and Suggestions for Authors

The paperEvaluation of dry eye treatment with Sodium hyaluronate and dexpanthenol containing eye-drops on ocular surface improvement after cataract surgery

conducted by Köppe et al. provides information regarding the approach to dry eye after cataract surgery.

The subject is relevant for research, because it brings new elements regarding the approach to dry eye after cataract surgery.

Standardization of treatment using HC increased the quality of life by reducing subjective symptoms and improving the stability of the tear film, 5 weeks after cataract surgery.

The evaluation made by patients and doctors indicated a high effectiveness and good tolerability.

The formulated conclusions are in accordance with the obtained results, based on the proposed study objectives.

The references are recent and relevant to the information presented.

The study is very interesting, but to improve the quality of the work, some changes should be made throughout the document.

1. Line 7: the repetition will be corrected.

2. PP population: to mention the meaning of the abbreviation, at the first mention in the text (in the abstract).

3. ITT population: idem.

4. Lines 119-120. Please, correct the terms used regarding administration, in English: "...and IsoptoMax eye ointment (Novartis 119 Pharma GmbH, Nuremberg, Germany) 1 × 1/nocte for 1 week).

5. I recommend changing the term "Graph" to "Figure"

6. Table 4: OD-oculus dexter is the Latin abbreviation for "right eye". Please, use the same terms.

7. To correct certain drafting errors, including in the case of figures.

Author Response

Thank you very much for your insightful comments. 

We ammended the manuscript in all your points accordingly. 

Thank you very much. 

Reviewer 3 Report (New Reviewer)

Comments and Suggestions for Authors

In the present manuscript, the authors assess if the employment of a specific teardrop may enhance the dry eye status of participants who underwent cataract surgery. While the issue is interesting, the study has some shortcomings that underscore the scientific value of the document. References in brackets should be positioned before the dot, not after (e.g., "post cataract surgery [1]. Dasgupta and Gupta").

The use of acronyms must be consistent. DED should be used everywhere once it is created. OS is employed sometimes and was never quoted. ITT was quoted more than once (it should only be quoted the first time it appears). The authors may revise the entire manuscript. In addition, there is a high number of acronyms, which sometimes make readability complicated. I recommend minimizing this issue. Also, figure and table legends should be independently written; therefore, acronyms should be included there.

The entire manuscript requires a revision of the presentation since line breaks often lack coherence. Additionally, punctuation should be reviewed. Similarly, tables and figures should adhere to the journal's guidelines for authors. In the present form, there is no consistency in presentation between them, which undermines the scientific presentation of the manuscript.

The entire manuscript lacks references for most statements and requires a comprehensive revision. Moreover, there are typos or inconsistencies regarding referencing systems in the text (e.g., lines 298 or 301) that require correction. Even the list of references requires revision, as different citation styles and systems are employed, indicating a lack of attention by the authors or careful revision before submission.

Methodology and sample size should be reviewed. Readers should not have to constantly refer back and forth to understand how the study was conducted and how many participants were recruited and finally included. There are redundancies and repeated information in different sections, making it difficult to discern the key points. I recommend creating a scheme or using bullet points to delineate the scientific impact.

The study needs a rationale behind the tests employed both for recruitment and as primary study goals. This is a major concern that should be carefully explained and justified in the introduction.

Please find below some examples (though there are more) that may help enhance the manuscript regarding the aforementioned recommendations:

Line 53: A reference is needed for this sentence (e.g., PMID: 35249333, PMID: 34545606).

Lines 54-61: Please revise. There appears to be a problem with line indentation.

Line 57: Please explain the acronym "OS" (I suppose it stands for "Ocular Surface").

Line 67: A reference is needed for this sentence (e.g., PMID: 33187860).

Line 76: A reference is needed (e.g., PMID: 37893592).

Lines 78-82: The presentation should be carefully revised.

Line 83: "well established"

Lines 104-106: All those cutoff criteria require references. Some of them seem a bit confusing regarding their usual employment (e.g., Why was OSDI set at a score equal to or higher than 16? The present study is based on the DEWS II principles, which recommend a 13-point score; PMID: 28736342).

Line 115: While logical, a reference is needed for the exclusion criteria (most of which are risk factors for dry eye disease by sources other than cataract surgery, e.g., PMID: 33485806, PMID: 37100346, PMID: 37087043). Additionally, the use of these strict criteria may be highlighted as a strength of the study in the discussion section.

Lines 124-130: The presentation of information should be improved.

Line 133: A reference is needed.

Lines 146–198: This entire section requires improvement in presentation.

Lines 218-219: Please revise the manuscript, as track changes seem to still be active during PDF elaboration. This issue is repeated several times (lines 225, 226, etc).

Author Response

Thank you very much for your invaluable and insightful comments. 

 In the present study we adhered to the recommendations of the DEWS II. We applied strict and high OSDI score of inclusion criteria to improve the strength of the study. We amended the entire manuscript in style and presentation. We included and clarified references where suggested.

Thank you very much again for your insightful comments. 

This manuscript is a resubmission of an earlier submission. The following is a list of the peer review reports and author responses from that submission.

Round 1

Reviewer 1 Report

Comments and Suggestions for Authors

I prefer to make the work design to have two groups, one with the usual postoperative treatment and the other group having the additional tested eye drops. Comparison between the two groups will be much valuable. I want to ask the authors and want to be included in the discussion, why you did not use OSDI scores in eye drop evaluation.

Comments on the Quality of English Language

Minor English editing is required.

Author Response

Dear Sir or Madame, 

Thank you very much for your invaluable comments. 

The purpose of the study was not to compare two eye drops but to proof the efficacy of hyaloron containing eye drops in postoperativ treatment of drye eye symptoms after cataract surgery. 

We did not use the OSDI scores in terms of eye drop evaluation because we rather used it as a screening tool to identify patients with dry eye disease and therefore to be suitable to be included in the study. We then used objective and not self reported measurement tools to evalauate the eficacy of the eye drops. 

Thank you very much again. 

Reviewer 2 Report

Comments and Suggestions for Authors

In this manuscript, authors have explored the efficacy of Sodium hyaluronate and 2 dexpanthenol containing eye-drops in improving the dry eye condition in subjects after cataract surgery. In total 39 study subjects were included for intention-to-treat population and 36 individuals for per-protocol analysis which satisfies the required sample size of 35 for a statistical 184 power of 95%. Except Schirmer test, the treatment is found to be effective as shown through increased TBUT and improved CDVA and corneal staining with minimal adverse events. Results presented justifies the conclusion made for HYLO CARE as an effective approach to improve dry eye symptoms in subjects with cataract surgery. 

There are grammatical errors in the manuscript and authors should correct it before resubmission.

Figure 1: What is the statistical significance between D0 and D35? Authors should depict that in the figure itself.

Line 230: “(as well as to all other trial time points). Mean TBUT doesn’t seems to be increased at D1 and D7 in comparison to D0.

Figure 2: Similar to fig 1. authors should present the difference between D0 and D35 is statistical significant.

Comments on the Quality of English Language

There are grammatical errors in the manuscript and authors should correct it before resubmission.

Author Response

Dear Sir or Madam, 

Thank you very much much for your insightful comments. 

We added the statistical significance in the Figure legends. 

Thank you very much again.